# Muscle strengthening in individuals with Amyotrophic Lateral Sclerosis: a systematic review with meta-analyses

Aline Alves de Souza[1,2]*, Stephano Tomaz da Silva[1,2],
Amanda Mayra Pereira Régis[1,2], Diogo Neres Aires[1,2], Karen de Medeiros Pondofe[1,2],
Luciana Protásio de Melo[2], Ricardo Alexsandro de Medeiros Valentim[2,3],
Ana Raquel Rodrigues Lindquist[1,2], Lorenna Raquel Dantas de Macedo[1],
Tatiana Souza Ribeiro[1,2]

1 Department of Physical Therapy, Federal University of Rio Grande do Norte, Natal, RN, Brazil,
2 Laboratory for Technological Innovation in Health, Federal University of Rio Grande do Norte, Natal, RN, Brazil, 3 Department of Biomedical Engineering, Federal University of Rio Grande do Norte, Natal, RN, Brazil

☯ These authors contributed equally to this work.
* alvss.aline@gmail.com

## Abstract

Despite the observed benefits of properly prescribed exercises for people with Amyotrophic Lateral Sclerosis (ALS), the scarcity of studies and lack of consensus on the effects of muscle-strengthening exercises on this population has a negative impact on their rehabilitation. This study aimed to evaluate the effects of muscle-strengthening interventions in individuals with ALS. This systematic review of intervention studies included clinical trials that performed non-respiratory muscle strengthening in people with ALS compared to non-strengthening interventions, usual care, or placebo. Such studies were obtained from the MEDLINE, EMBASE, Cochrane Library, SPORTDiscus, and Physiotherapy Evidence Database databases, with no language or publication date restrictions. The outcomes considered were peripheral muscle strength, functionality, fatigue, and adverse events. The Physiotherapy Evidence Database scale was used to analyze the risk of bias, while the Grading of Recommendations Assessment, Development and Evaluation system was used to evaluate the quality of the evidence. Searches were conducted in October 2023 and eight studies were included, totaling 296 individuals. Seven of the eight studies showed superiority of the experimental intervention over the control, but this was not supported in the meta-analyses. Small sample size and high heterogeneity in the primary studies contributed significantly to the low quality of the evidence. There was no evidence of the superiority of interventions for muscle strengthening compared to interventions not aimed at strengthening, usual care, or placebo in terms of the outcomes analyzed immediately after the intervention. The quality of the evidence ranged from low to very low. Five of the studies evaluated adverse events, without reporting serious events. Interventions for muscle strengthening did not prove to be more effective when compared to the control group in the short term nor seem to produce serious adverse events. The low quality of the evidence indicates the need for studies with greater methodological rigor in this population, to more assertively assess the impacts of this intervention over the short, medium, and long term.

**Data availability statement:** All relevant data are within the manuscript and its Supporting Information files. The minimal data set can be found in the S4 file.

**Funding:** This study was supported by the Coordenação de Aperfeiçoamento de Pessoal de Nível Superior – Brazil (CAPES) – Finance Code 001, and by the National Council for Scientific and Technological Development and Ministry of Health, through a Decentralized Execution Term (TED 132/2018). The funders had no role in study design, data collection and analysis, decision to publish, or preparation of the manuscript.

**Competing interests:** The authors ha.ve declared that no competing interests exist.

## Introduction

Amyotrophic Lateral Sclerosis (ALS) is a progressive neurodegenerative disease caused by the degeneration of upper and lower motor neurons in the cerebral cortex, brainstem, and spinal cord [1]. Considered a rare disease, the worldwide prevalence is around 1.0–11.3 cases per 100,000 people [2], and the incidence is 0.6 to 3.8 per 100,000 people annually [3], affecting people with an average age of 60 years [2].

Genetic and environmental factors are involved in the onset of ALS [4], which has heterogeneous clinical characteristics, with signs and symptoms that usually vary according to the form of presentation of the disease. Thus, ALS can be classified as spinal onset—when there is initial involvement of the upper and/or lower limbs, resulting in a predominant picture of muscle weakness, atrophy, fatigue, fasciculation, and others; or bulbar onset—when degeneration begins in the motor neurons of the brainstem, with weakness of the facial muscles, dysarthria, dysphagia, and impairment of the respiratory muscles, which can lead to death due to ventilatory failure [5].

Since there is no curative drug treatment [6], and considering the complexity of this condition, it is essential to work with a multi-professional team capable of meeting the demands throughout the disease course. Physiotherapy makes it possible to manage impairments that impact functional independence and quality of life, as well as delaying the occurrence of future complications [7,8].

One of the main components of the physiotherapy approach for individuals with ALS is physical exercise. When properly prescribed, exercise has physical and psychological benefits for this population, regardless of the stage of the disease [9]. Current evidence shows that aerobic exercise can increase the functional capacity of individuals with ALS [10]. In addition, therapeutic exercises have been found to reduce muscle deterioration and make it easier for these people to carry out day-to-day activities [10].

Due to the progressive muscle weakness inherent in ALS, individuals may experience a process known as overuse, which is generated by excessive demands on the skeletal muscles [9]. Thus, there is frequent discouragement of the prescription and execution of strengthening exercises as a way to prevent overuse, which would trigger greater muscle weakness [10]. However, it is worth considering that too little physical activity—a process known as disuse—is not beneficial, as it leads to hypotrophy and muscle weakness, as well as deterioration of the individual's clinical condition [9].

Despite the known benefits of physical exercise, most of the available evidence for this population addresses exercise programs with a predominance of aerobic modalities, passive exercises, and stretching, with little focus on muscle-strengthening exercises [11]. This can have a negative impact on the quality of the rehabilitation offered to individuals with ALS, as it limits the use of scientifically based procedures.

Considering the controversial evidence in the literature and the scarcity of studies focusing on muscle strengthening in this population, this study aims to analyze the existing evidence regarding the effects of physiotherapeutic interventions for muscle strengthening in the short, medium, and long term, verifying the effects of these interventions on the outcomes of peripheral muscle strength, functionality, and fatigue in individuals with ALS.

## Materials and methods

This systematic review was prepared according to the *Preferred Reporting Items for Systematic Reviews and Meta-Analyses* (PRISMA) guidelines [12]. The protocol for this work was registered in the *Open Science Framework* (OSF) (https://doi.org/10.17605/OSF.IO/ZFA7H) and published in a scientific journal [13].

The studies were included in the review following the PICOTS strategy (Population, Interventions, Comparators, Outcomes, Time frame, and Study design/Settings) [14].

## Types of study

Randomized, quasi-randomized, or non-randomized clinical trials were included, which addressed the effects of motor interventions for muscle strengthening in individuals with ALS. Crossover studies were included when the data from the interventions (control and experimental) were presented separately.

## Participants

Individuals with a confirmed or probable clinical diagnosis of ALS, according to the *El Escorial* [15] criteria, of both sexes, aged 18 or over, and who had undergone motor interventions to strengthen their muscles in the respective primary studies were included. For studies involving several pathologies, only studies in which the data of participants with ALS were made available separately were included.

## Experimental interventions and comparators

Regarding the experimental intervention, all physiotherapeutic practices aimed at strengthening the skeletal muscles of the upper limbs, lower limbs, face, cervical region and/or trunk were considered, such as therapeutic exercises and electrostimulation, for example. Studies with physiotherapeutic interventions only to strengthen the respiratory muscles were not included.

For the comparators (control groups), the following were considered.

- Any intervention (physiotherapeutic or otherwise) without the aim of strengthening muscles;

- Minimal intervention (guidelines, booklets, etc.), with no objective of muscle strengthening;

- Placebo or no intervention

## Outcomes

**Primary.** Peripheral muscle strength was considered the primary outcome and was measured using the Manual Strength Test (MST) [16], the Muscle Strength Grading Scale [16], an isokinetic dynamometer [17], or a hand-held dynamometer [18].

**Secondary.** Were considered secondary outcomes: (i) muscle activation, assessed by electromyography (EMG); (ii) functionality, assessed by the Amyotrophic Lateral Sclerosis Functional Rating Scale-Revised (ALSFRS-r) [19]; (iii) fatigue, assessed by the Fatigue Severity Scale (FSS) [20] and the Borg Rating Of Perceived Exertion scale [21]; and (iv) adverse events, such as pain, discomfort, fatigue or others reported in the study, descriptively or using specific evaluation instruments.

## Evaluation moment

The outcome data was extracted at the time of the initial assessment (baseline), immediately after the end of the intervention (immediate effects) and the follow-up assessment, referring to the second measurement, carried out up to 3 months after the end of the intervention (medium-term effects) or after three months (long-term effects).

### Research methods and strategies

The studies were obtained from five scientific health databases: Medical Literature Analysis and Retrieval System Online (MEDLINE), EMBASE, Cochrane Library (CENTRAL), SPORTDiscus, and Physiotherapy Evidence Database (PEDro). The gray literature was also researched by conducting searches of clinical trial registers: USA National Institutes of Health Ongoing, and Trials Register ClinicalTrials.gov, as well as from the reference lists of the included studies. The searches were carried out without restricting language, geographical area, or date of publication. The search strategy (see S1 File) was developed using MEDLINE and adapted for the other databases, using descriptors in English referring to aspects of the condition (ALS), outcomes, and types of study. The searches were conducted in October 2023.

### Data screening and extraction

Two independent reviewers (AS and SS) analyzed the titles and abstracts of the references and excluded irrelevant studies using the Rayyan® software (Intelligent Systematic Review) [22]. Duplicate studies were removed, with only one copy being recorded. When the duplication was not explicit and more than one study was included with the same sample, the authors were contacted to confirm the duplication. If duplication was confirmed, the most recent study or the one with the most complete data presentation was considered. Then, potential eligible studies were read in full and selected according to the inclusion criteria. Any disagreement was resolved through discussion or consultation with a third author (DA) for a judgment. The process was recorded on a PRISMA flowchart [12].

The same reviewers who analyzed the titles extracted the data from the included studies using a previously defined extraction form (See S2 File). Data extracted include the following: name of the first author, year of study publication, type of study, sample size, sample characteristics, outcome measures, description of interventions, and results. In the event of missing or uncertain data, the author of the study was contacted for clarification.

### Risk of bias assessment

The Physiotherapy Evidence Database Scale (PEDro) [23] was used to analyze the risk of bias in the studies. This scale consists of 11 criteria, which assess the methodological rigor of clinical trials and determine the quality of the study based on a score ranging from 0–10, where the higher the score of the study, the better the methodological rigor. Scores of five or above determine adequate methodological rigor [23].

### Data analysis and processing

Statistical analyses were carried out using Review Manager® software [24]. For continuous outcomes, the weighted mean difference was selected when the measurement tools and indicator units were the same. The standardized mean difference was selected when the tools and units of measurement indicators were different. All items above have been represented by the effect value and 95% confidence interval. For the dichotomous outcomes (adverse events), the Odds Ratio (OR) was used to measure the effects of the treatment, with a 95% confidence interval.

### Heterogeneity

Heterogeneity was determined by the values of $\chi^2$ and $I^2$. If P $\geq$ 0,1, $I^2 \leq$ 50%, which indicates low heterogeneity, the random model was used for the meta-analysis cases. If $P <$ 0,1, $I^2 >$ 50%, which indicates heterogeneity between the studies, the source of heterogeneity was explored employing subgroup analysis [25].

## Meta-analyses

The meta-analyses were conducted using the *Review Manager*® software [24]. When possible, studies were grouped into meta-analyses to compare the effects of the following:

i) Physiotherapeutic interventions aimed at muscle strengthening versus placebo;

ii) Physiotherapeutic interventions aimed at muscle strengthening versus no therapy;

iii) Physiotherapeutic interventions aimed at muscle strengthening versus any other intervention not aimed at muscle strengthening;

iv) Physiotherapeutic interventions aimed at muscle strengthening versus educational and guidance programs.

## Subgroup analyses

The following subgroup analyses have been predefined to be executed when possible:

i) Type of onset: spinal or bulbar;

ii) Duration of illness: less than five years or more than five years;

iii) Age: under 60 and over 60;

iv) Type of treatment: strengthening performed manually or using instruments/equipment (shin pads, machines, elastic bands);

v) Treatment dose: 1–2 times a week and more than two times a week.

## Sensitivity analyses

Sensitivity analyses were considered when there was a suspicion of missing data that could introduce important biases, and also to assess the heterogeneity caused by peripheral studies. In addition, studies with low scores in the risk of bias assessment (<5) were considered for exclusion in the sensitivity analysis.

## Assessing the quality of evidence

The five considerations of the Grading of Recommendations Assessment, Development and Evaluation (GRADE) were used [26]: study limitations, inconsistency, imprecision, indirect evidence, and publication bias. Decisions to downgrade or upgrade the quality of evidence were justified with footnotes and comments to aid the reader's understanding when necessary. According to the GRADE, the quality of the evidence was classified into strata, ranging from high, moderate, low, or very low.

# Results

## Study selection

After the searches, 10,921 studies were found. Next, duplicate studies were selected and removed, and 24 potentially eligible references were identified. After thoroughly reading the texts, eight studies were included in the review. The search results are summarized in the flowchart below (Fig 1).

## Characteristics of the studies

The references included consisted mostly of randomized clinical trials, except for two references, two of which were non-randomized clinical trials (Jensen, 2017; Kitano, 2018). A total

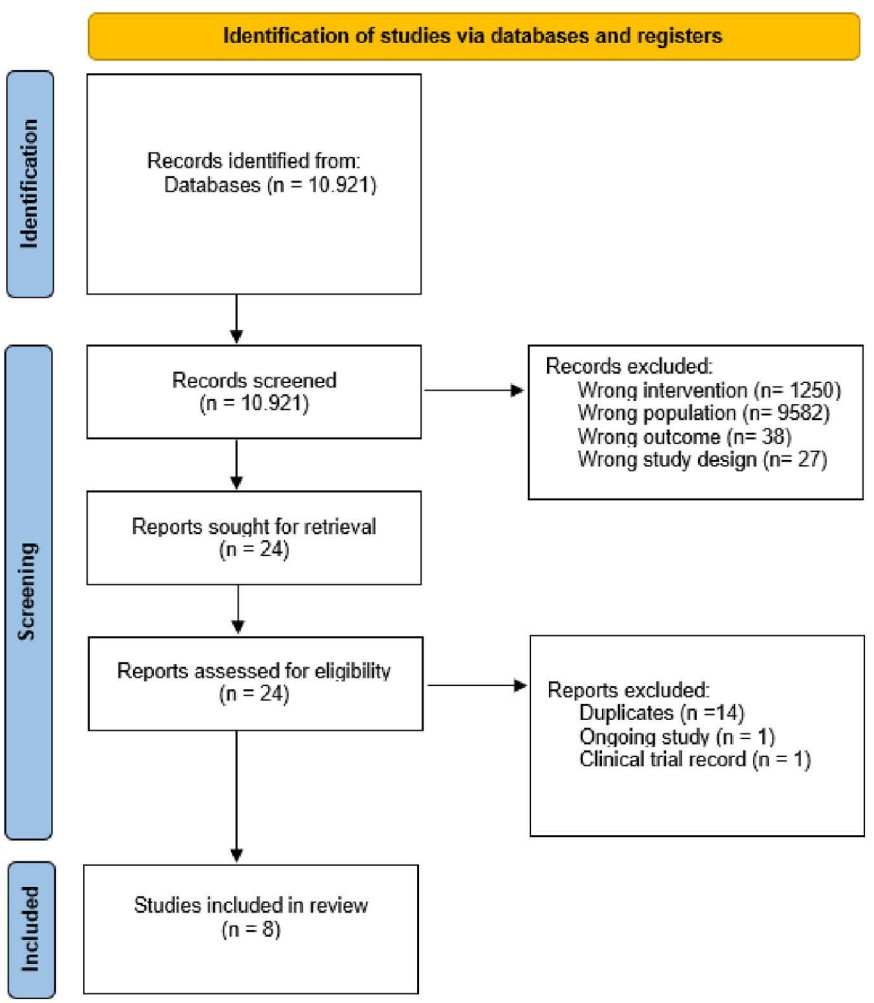

**Fig 1. PRISMA flowchart describing the selection of studies.**

of 296 individuals participated in the eight studies, 94 of whom were female and 202 males. According to the form of presentation of the disease, 194 had spinal-onset ALS, and 102 had bulbar-onset ALS. Table 1 shows the characteristics of the included studies.

**Table 1. Characteristics of the included studies (n = 8).**

| First author and year | Country | Total sample size | Type of study | Location |
|---|---|---|---|---|
| **Drory, 2001** | Israel | 25 | Randomized clinical trial | Participant's domicile |
| **Ferri, 2019** | Italy | 16 | Randomized clinical trial | Clinic |
| **Groenestijn, 2019** | Netherlands | 57 | Multicentre randomized clinical trial | Participant's domicile and clinic |
| **Jensen, 2017** | Denmark | 5 | Non-randomised clinical trial | Medical centre |
| **Kalron, 2021** | Israel | 28 | Randomized clinical trial | Medical centre |
| **Kitano, 2018** | Japan | 105 | Non-randomized clinical trial | Participant's domicile |
| **Merico, 2018** | Italy | 38 | Pilot randomized clinical trial | Clinic |
| **Musarò, 2019** | Italy | 22 | Randomized clinical trial | Clinic |

## Outcome measures

The outcome of peripheral muscle strength was assessed in six studies (Drory, 2001; Ferri, 2019; Groenestijn, 2019; Jensen, 2017; Kalron, 2021; Merico, 2018), using the TMS, Muscle Strength Grading Scale, isokinetic dynamometer or hand-held dynamometer as the assessment tool. Functionality was assessed in six studies (Drory, 2011; Ferri, 2019; Groenestijn, 2019; Jensen, 2017; Kalron, 2021; Kitano, 2018), all using the ALSFRS-r as an assessment tool. Fatigue was assessed in four studies, using the Fatigue Severity Scale (Drory, 2001; Groenestijn, 2019; Kalron, 2019; Merico, 2018) and the fatigue subscale of the Checklist of Individual Strength (CIS) [27] to assess this outcome.

## Experimental interventions and comparators

The physiotherapy interventions used involved resistance, aerobic, and postural balance exercises combined with muscle strengthening. In addition, interventions using neuromuscular electrical stimulation were also applied. For comparison purposes, the control groups in the studies used maintenance of usual activities, placebo, stretching, and manual therapy. The data on the interventions and comparators are described in Table 2.

## Risk of study bias

Of the eight studies included, three (Drory, 2001; Kitano, 2018; Jensen, 2017) did not present adequate methodological rigor, which resulted in a low score on the PEDro scale. Of these, two studies (Kitano, 2018; Jensen, 2017) did not have adequate randomization. Concealed allocation was met in two studies (Kalron, 2021; Groenestijn, 2019). Two studies (Musarò, 2019; Jensen, 2017) did not have similar groups at the start of the study. Participants and therapists were not blinded in any of the eight studies. The outcome assessors were blinded in four studies (Merico, 2018; Musarò, 2019; Kalron, 2021; Groenestijn, 2019). In three studies (Drory, 2001; Groenestijn, 2019; Kitano, 2018), there was more than a 15% drop-out rate among participants. The intention-to-treat analysis was carried out in six studies (Merico, 2019; Ferri 2019; Musarò, 2019; Groenestijn, 2019; Kitano, 2018; Jensen, 2017). Statistical comparison between the groups was carried out in all studies. Point measures and variability data were presented in five studies (Merico, 2019; Ferri, 2019; Kalron, 2021; Groenestijn, 2019; Kitano, 2018). The items and scores for risk of bias are described in Table 3.

## Quantitative data synthesis – meta-analyses

Meta-analyses were carried out evaluating the outcomes of peripheral muscle strength, functionality, and fatigue, taking into account the immediate effects observed once the intervention had ended since none of the included studies presented data on medium and long-term effects. The study by Musarò (2019) presented the results only in figures, and it was not possible to extract numerical data on the evaluations for inclusion in the meta-analyses. The author in question was contacted, but no response has been received to date.

It was not possible to carry out meta-analyses on adverse events. No serious adverse events were reported in four of the included studies (Ferri, 2019; Kitano, 2018; Groenestijn, 2019; Kalron, 2021). Three studies (Drory, 2011; Jensen, 2017; Merico, 2018) did not consider the evaluation of adverse events; and one of the studies (Musarò, 2019) did not present data on the adverse events evaluated. The study by Groenestijn (2019) described the occurrence of five adverse events in a total of 614 training sessions (0.8 percent), where one individual experienced myalgia, two experienced increased fasciculation, and two experienced nocturnal cramps [27].

**Peripheral muscle strength.**  Six studies (n = 134) assessed peripheral muscle strength using the TMS (Drory, 2001), isokinetic dynamometer (Merico, 2018), hand-held

Table 2. Experimental interventions and comparators of the included studies (n = 8).

| Study | Intervention group (experimental) | Control group (comparator) | Frequency and duration of interventions |
|---|---|---|---|
| **Drory, 2001** | Participants were given a list of exercises by a physiotherapist involving muscle groups of the four limbs and trunk, considering the general health status, neurological status, and level of physical performance of each participant. The main aim of the exercise program was to improve muscular endurance, using a moderate load but causing the muscles to change significantly in length. The program was demonstrated to each patient individually and reviewed at each clinic visit. | The patients were instructed not to carry out any physical activity other than their activities of daily living. | The program was designed to last 15 minutes and should be done twice a day at home for three months. |
| **Ferri, 2019** | It consisted of aerobic, resistance, balance, and stretching exercises, organized as follows: 15 minutes on a cycle ergometer at an intensity corresponding to 80% between the baseline and the gas exchange threshold calculated during a cardiopulmonary ergometric test; strength exercises at an intensity of 60% of 1RM; 3 sets of 10 repetitions (with a 2-minute break between each set) were carried out for the upper limbs: biceps flexion and shoulder abduction; and lower limbs: squat, plantar flexion, and knee extension (isotonic machine); with the use of dumbbells and Thera-Band for bench press and seated rowing exercises. Strength exercises were performed alternately throughout the week; the eccentric phase of the exercises was avoided; 10 minutes of proprioceptive exercises, mostly performed on the BOSU Pro balance trainer; 10 minutes of upper and lower limb stretching exercises performed on a Pancafit. | Participants received manual therapy once a week or fortnight and were instructed to maintain activities of daily living. | 60-minute sessions, 3 times a week for 12 weeks. |
| **Groenestijn, 2019** | (1) Home training consisted of individualized aerobic exercise on a cycle ergometer and a stepboard. Training intensity was gradually increased from 50 percent (moderate) to 75 percent (vigorous) of the HRF and the duration of the training was gradually increased from 20 to 35 minutes/session. (2) The 1-hour supervised individual training sessions consisted of work stations; each session was divided into a 5-minute warm-up, 30 minutes of individually adapted aerobic exercises (cycle ergometer, step board, and treadmill), 20 minutes of muscle strengthening exercises (quadriceps, biceps, and triceps) and a 5-minute cool-down. The intensity of the aerobic exercise training was gradually increased from 50% to 75% of the HRF, and the intensity of the muscle strengthening exercise training was gradually increased from 40% to 50% of the maximum strength of the different muscle groups (quadriceps, biceps, and triceps). Each exercise was repeated 10–15 times. | All participants received neuropalliative care from multidisciplinary secondary care teams. These teams consisted of a rehabilitation medicine consultant, an occupational therapist, a physiotherapist, a speech therapist, a nutritionist, a social worker, a psychologist, and consultant physicians. No allocation group was restricted in their daily activities. | The aerobic exercise therapy group lasted 16 weeks and consisted of (1) a home-based training program twice a week and (2) an individual training session once a week. |
| **Jensen, 2017** | The protocol consisted of resistance exercises focused on the upper and lower body (leg press, knee extension, knee flexion, heel raise, seated row, bench press, shoulder development, abdominal crunches, and back extension) and included an alternating program using 6 exercises each session. The training was carried out in small groups supervised by two experienced physical trainers and each session began with a 5-minute warm-up on a stationary bike. The initial two weeks were used for familiarisation with 3 sets of 12–15 RM and, from then on, the load progressed throughout the study, ending with 2 sets of 5–6 RM (using the 5 RM test). Continuous adjustments to the exercises were made to ensure optimal conditions. | The participants only maintained their usual daily activities. | The exercise protocol should be carried out 2–3 times per week on non-consecutive days. |
| **Kalron, 2021** | Each session consisted of three modes: (1) aerobic training through recumbent cycling at 40–60% of HRF; (2) flexibility obtained through stretching exercises and passive range of motion; and (3) strength training through functional exercises based on the individual's body weight. The strength exercises were performed in different body positions (e.g., sitting, supine, or with hands and legs on the floor), focusing on the large muscle groups of the trunk and upper and lower limbs. Examples include traditional exercises like squats, planks, lunges, elbow flexions, and pelvic lifts. For each exercise, the aim was to perform 1–2 sets, each consisting of 8–12 repetitions. The rest time was according to the individual's fatigue. | The participants were instructed to perform basic stretching exercises for their upper and lower limbs at home, with the help of their carer or family member. The stretching exercises were presented in a booklet with illustrations and a short text emphasizing the correct execution technique. Adherence to the exercises was monitored by a self-report diary and by a physiotherapist who contacted the patients every fortnight. | For the experimental group: the program included 24 sessions spread over 12 weeks (2 sessions per week), each lasting 50–60 minutes (20–30 min of aerobics, 10 min of flexibility, and 20 min of strengthening). For the control group: the participants were instructed to perform the stretching exercise for 20 minutes, 5 times a week, for 12 weeks. |

*(Continued)*

**Table 2.** (Continued)

| Study | Intervention group (experimental) | Control group (comparator) | Frequency and duration of interventions |
|---|---|---|---|
| **Kitano, 2018** | The program consisted of home exercises made up of the following 7 exercises: (1) two types of strengthening and stretching exercises for the upper limbs and trunk; (2) three types for the lower limbs and trunk; (3) two types of functional exercises for activities of daily living, such as rolling over and getting up from a chair. For the strengthening exercises, the resistance used was body weight in antigravity positions and without any equipment. | Participants were given exercises supervised by a physiotherapist for six months. | The physiotherapist prescribed exercise frequency and repetitions for each patient individually, based on an assessment of the patient's condition and functioning. |
| **Merico, 2018** | Progressive individualized treatment for muscle strengthening and aerobic endurance, under the supervision of two physiotherapists. The submaximal isometric contraction was used on muscles with a score of 3, 4, or 4 on the MRC scale. Three repetitions were performed bilaterally and were carried out in a time interval determined by calculating 80 percent of the maximum contraction time, with 30 seconds of rest. The aerobic exercises, cycle ergometry, arm-leg ergometry, and/or walking, were carried out at a sub-maximal intensity of 65% of the age-adjusted heart rate and lasted 15–20 minutes. All participants underwent speech therapy, occupational therapy, and psychological therapy based on their individual clinical and functional profiles. | One-hour sessions of stretching, active mobilization, and general muscle strengthening; the latter treatment was defined within the limits of the fatigue reported by the patient and avoiding eccentric and concentric contractions. All patients underwent speech therapy, occupational therapy, and psychological therapy, based on their individual clinical and functional profiles. | The series of exercises was carried out daily for 5 weeks and, in general, each exercise session lasted about an hour. |
| **Musarò, 2019** | Real ENMS was performed on the right arm and simulated neuromuscular electrical stimulation on the left arm in the same daily session | Real ENMS were performed on the left arm and simulated ENMS on the right arm | All participants received daily repetitive real and simulated ENMS sessions, five days a week for two consecutive weeks. |

Key ENMS: Electrical Neuromuscular Stimulation; HRF: Heart Rate Reserve; MRC: Medical Research Council; RM: Repetition Maximum.

**Table 3. Physiotherapy Evidence Database (PEDro) scale scores for risk of bias of included studies (n = 8).**

| Studies | PEDro scale items | | | | | | | | | | | | Methodological rigor |
|---|---|---|---|---|---|---|---|---|---|---|---|---|---|
| | 1* | 2 | 3 | 4 | 5 | 6 | 7 | 8 | 9 | 10 | 11 | Total | |
| **Merico, 2018** | | + | − | + | − | − | + | + | + | + | + | 7 | Adequate |
| **Drory, 2001** | | + | − | + | − | − | − | − | − | + | − | 3 | Inadequate |
| **Ferri, 2019** | | + | − | + | − | − | − | + | + | + | + | 6 | Adequate |
| **Musarò, 2019** | | + | − | − | − | − | + | + | + | + | − | 5 | Adequate |
| **Kalron, 2021** | | + | + | + | − | − | + | + | − | + | + | 7 | Adequate |
| **Groenestijn, 2019** | | + | + | + | − | − | + | − | + | + | + | 7 | Adequate |
| **Kitano, 2018** | | − | − | + | − | − | − | − | + | + | + | 4 | Inadequate |
| **Jensen, 2017** | | − | − | − | − | − | − | + | + | + | − | 3 | Inadequate |

**Key:** (+): criterion met, (-): criterion not met.

*Item not included in the average score.

Items:

1. Eligibility criteria specified.

2. Random allocation.

3. Hidden allocation.

4. Similar groups at the beginning of study.

5. Blinding of subjects.

6. Blinding the therapist.

7. Blinding the evaluator.

8. Less than 15% dropout rate.

9. Intention-to-treat analysis.

10. Statistical comparisons between groups.

11. Point measurements and variability data.

dynamometer (Groenestijn, 2019), 1-repetition maximum test (Ferri, 2019), 30-second sit-and-stand test (Jensen, 2017) and 5-second sit-and-stand test (Kalron, 2021). The data show that there was no difference between the effect of the experimental interventions compared to the control on peripheral muscle strength (pooled mean difference = 0.28; 95% CI -0.47 to 1.03; P = 0.46; $I^2$ = 76%).

1.1    Peripheral muscle strength (Fig 2)

**Functionality.** Six studies (n = 195) assessed functionality using the revised ALSFRS (ALSFRS- r). The data show that there was no difference between the effect of the experimental interventions and the control on the functionality outcome (pooled mean difference = 0.67; 95% CI -0.07 to 1.40; P = 0.07; $I^2$ = 74%).

1.2    Functionality (Fig 3)

**Fatigue.** Four studies (n = 118) assessed the degree of fatigue using the FHS (Drory, 2001; Kalron, 2021; Merico, 2018) and the CIS fatigue subscale (Groenestijin, 2019). The data show that there was no difference between the interventions and the control for the fatigue outcome (grouped mean difference = 1.54; 95% IC -1.02 to 4.11; P = 0.24; $I^2$ = 96%).

1.3    Fatigue (Fig 4)

## Subgroup analyses

Such analyses were carried out for type of disease onset, age, type of treatment, and treatment dose. It was not possible to group the studies according to the duration of the disease due to

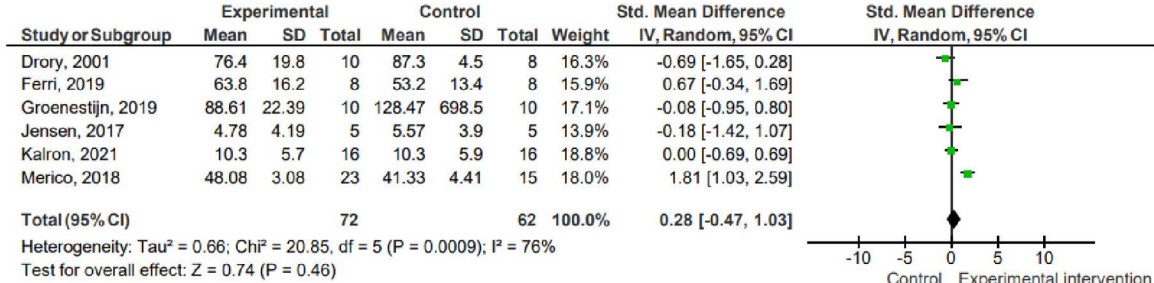

**Fig 2. Effects on peripheral muscle strength.** Key: SD – Standard deviation; MD – Mean difference; CI – Confidence interval.

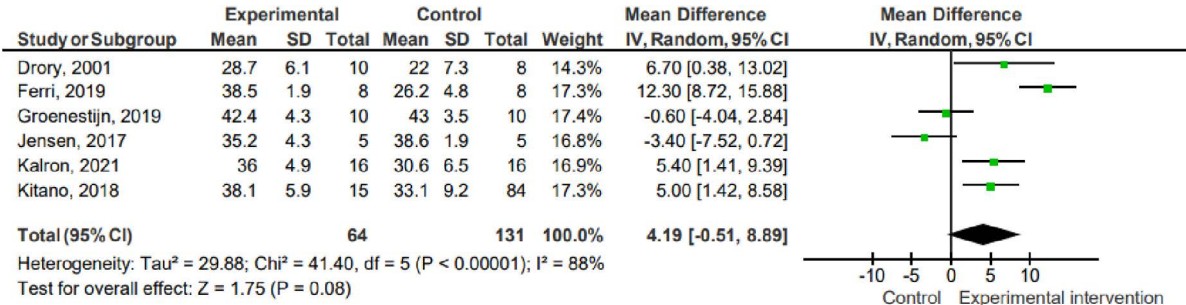

**Fig 3. Effects on functionality.** Key: SD – Standard deviation; MD – Mean difference; CI – Confidence interval.

*(Continued)*

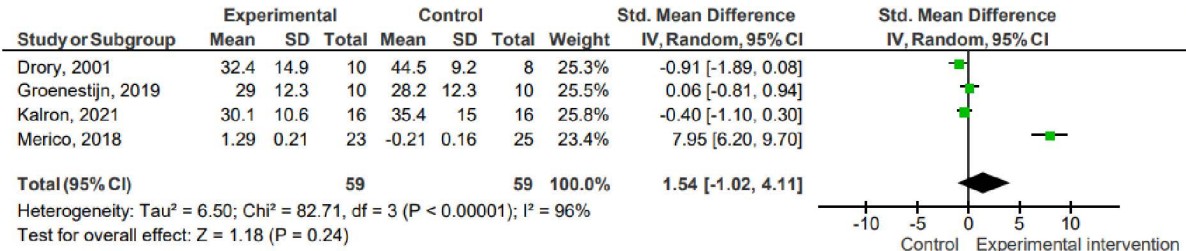

**Fig 4. Effects on fatigue.** Key: SD – Standard deviation; MD – Mean difference; CI – Confidence interval.

insufficient data on the time of onset of symptoms and time of diagnosis in the primary studies included.

**Weekly frequency.** In the subgroup analysis considering weekly frequency, the studies were allocated as follows: (1) frequency of two sessions per week (two studies, 50 participants); (2) frequency of more than two sessions per week (four studies, 84 participants). There was no statistical significance in favor of the intervention or control group in any subgroups. There was no inter-group statistical significance (P = 0.15; $I^2$ = 51.1%).

2.1    Weekly Frequency (Fig 5)

**Type of treatment.** In the subgroup analysis considering the type of treatment used, the studies were allocated as follows: (1) functional or bodyweight exercises (one study, 32 participants); (2) exercises using accessories such as dumbbells and elastic bands (three studies, 64 participants). There was no statistical significance in favor of the intervention or control group in any of the subgroups. There was no inter-group statistical significance (P = 0.22; $I^2$ = 34.8%).

2.2    Type of treatment (Fig 6)

**Age.** In the subgroup analysis considering the age of the participants, the studies were allocated as follows: (1) age under 60 (three studies, 66 participants); (2) age over 60 (three

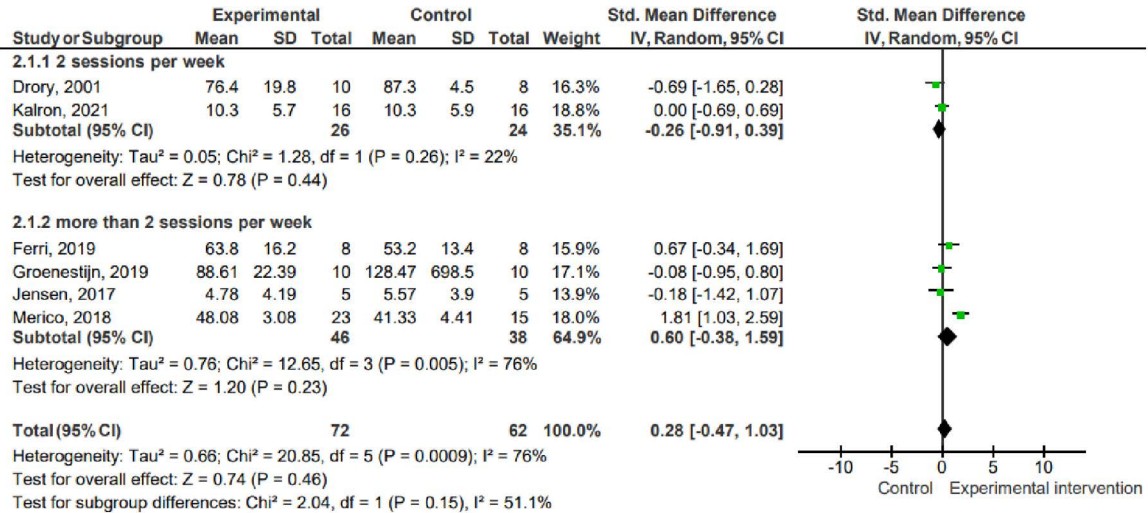

**Fig 5. Subgroup analysis: weekly frequency.** Key: SD – Standard deviation; MD – Mean difference; CI – Confidence interval.

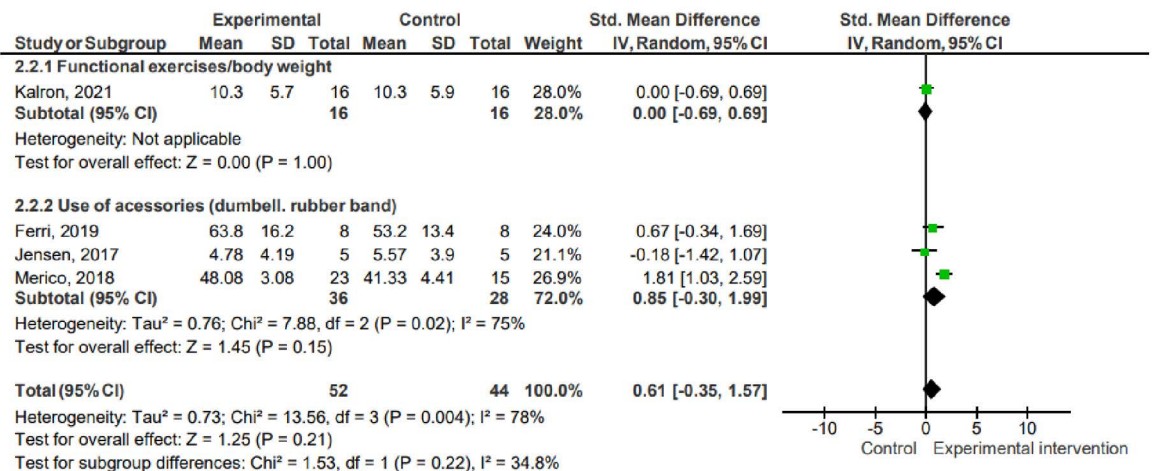

**Fig 6. Subgroup analysis: type of treatment.** Key: SD – Standard deviation; MD – Mean difference; CI – Confidence interval.

studies, 68 participants). There was no statistical significance in favor of the intervention or control group in any of the subgroups. There was no inter-group statistical significance (P = 0.55; I² = 0%).

2.3 Age (Fig 7)

## Sensitivity analysis

The sensitivity analysis for the primary outcome muscle strength was carried out considering the exclusion of studies capable of generating bias (Ferri, 2019; Groenestijn, 2019; Merico, 2018), as they did not have adequate methodological rigor (score < 5 on the PEDro scale). However, there was no change when the sensitivity analysis was carried out compared to the meta-analysis grouping all the studies together (Fig 8).

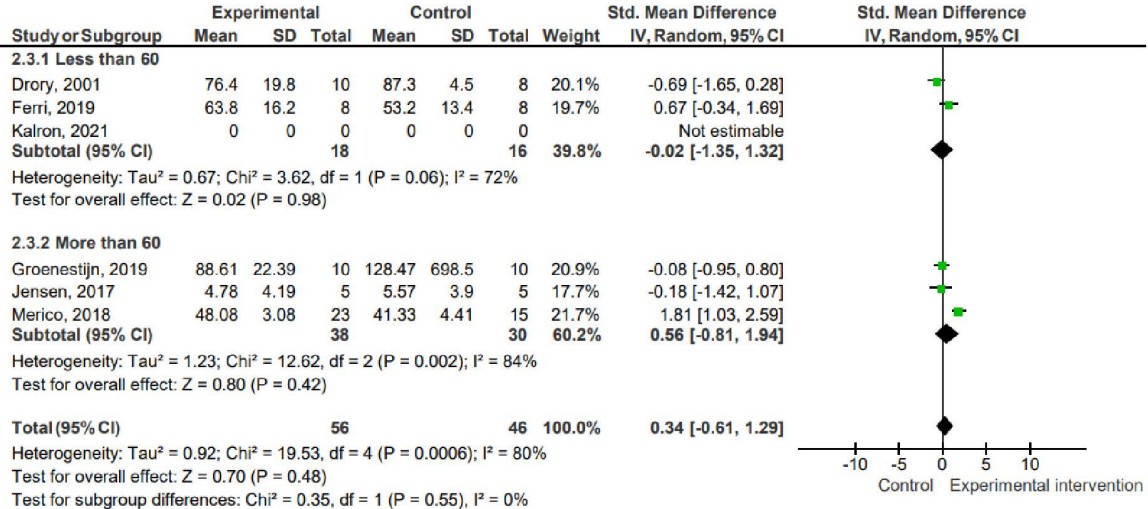

**Fig 7. Subgroup analysis: age.** Key: SD – Standard deviation; MD – Mean difference; CI – Confidence interval.

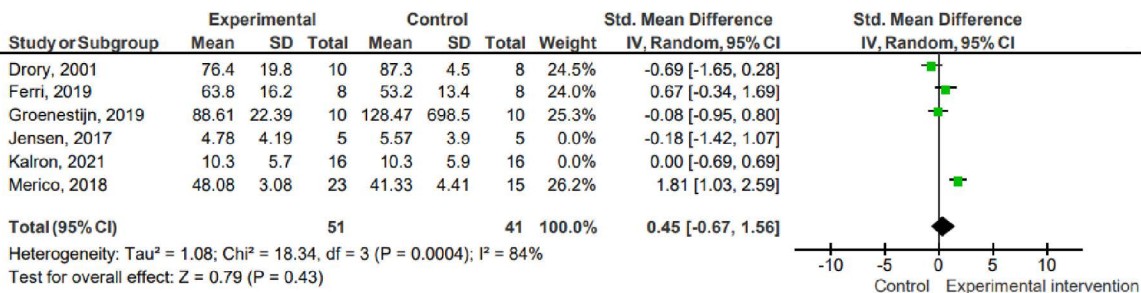

**Fig 8. Sensitivity analysis.** Key: SD – Standard deviation; MD – Mean difference; CI – Confidence interval.

## Evidence quality analysis

Three GRADE domains—risk of bias, inconsistency and imprecision—presented problems and were responsible for lowering the quality of evidence in the outcomes evaluated in the studies. The certainty of the evidence was classified as very low for the peripheral muscle strength and functionality outcomes, and as low for the fatigue outcome. The assessment of the certainty of the evidence and the reasons for the downgrade are reported in Table 4.

## Discussion

In this review, pieces of evidence present in the literature were evaluated to verify the effects of physiotherapeutic interventions for peripheral muscle strengthening in individuals with ALS. After searching the databases, eight studies were included, bringing 296 participants. Experimental interventions that consisted of strengthening exercises for the upper and lower limbs using elastic bands/halters, neuromuscular electrical stimulation, isotonic exercises, or using the body weight, were used as a means of comparison for control groups, in which the patients received a non-strengthening intervention placebo or maintenance of usual care.

**Table 4. Assessment of the quality of evidence based on the Grading of Recommendations Assessment, Development and Evaluation (GRADE).**

| Outcome | Anticipated absolute effects (95% CI) Risk with motor interventions | Relative effect (95% CI) | No. of participants (studies) | Certainty of evidence (GRADE) | Comments |
|---|---|---|---|---|---|
| Peripheral muscle strength assessed by the Manual Strength Test, isokinetic dynamometer or hand dynamometer | MD 1.56 upper (3.54 lower than 6.66 higher) | – | 134 (6 studies) | ⊕◯◯◯ Very low a,b,c | |
| Functionality assessed by the ALSFRS-r | MD 4.19 higher (0.51 lower to 8.89 higher) | – | 195 (6 studies) | ⊕◯◯◯ Very low a,b,c | The evidence was not in favor of motor interventions or control for functionality |
| Fatigue assessed by the Fatigue Severity Scale | MD 2.45 lower (8.45 lower to 3.56 higher) | – | 118 (4 studies) | ⊕⊕◯◯ Low b,c | |

Key CI: Confidence Interval; MD: Mean Difference; ALSFRS-r: Amyotrophic Lateral Sclerosis Functional Rating Scale - revised; GRADE: Grading of Recommendations Assessment, Development and Evaluation.

Rationale for downgrade:

a. Downgraded one level due to important classifications with inadequate methodological rigour.

b. Downgraded one level due to moderate or significant heterogeneity (>50%).

c. Downgraded one level due to small sample size (<400).

This study found that a descriptive analysis of the primary studies indicated the superiority of the experimental intervention compared to the control, however, the meta-analyses showed that physiotherapeutic interventions for peripheral muscle strengthening were not effective in individuals with ALS when compared to the control group immediately after the end of the intervention. Since the certainty of the evidence was considered low, there is limited confidence in the estimated effect of this comparison. The subgroup analyses regarding the weekly frequency of interventions, types of treatments carried out in the experimental group, as well as the ages of the participants, also did not prove to be in favor of either group (experimental or control).

In an individual analysis, of the eight studies incorporated in this review, seven showed favorable results for the experimental intervention when compared to the control (Drory, 2011; Ferri, 2019; Jensen, 2017; Kalron, 2021; Kitano, 2018; Merico, 2018; Musarò, 2019). The main results showed maintenance or improvement in functionality, fewer reports of fatigue, and an increase in muscle strength among the individuals who underwent the intervention. One of the studies (Groenestijin, 2019) did not observe any positive effects of adding the experimental intervention compared to the usual care provided to individuals with ALS. Despite the reported benefits, the high dropout rate of participants is consistently highlighted as a limitation in primary studies, generating considerable bias in the results presented. Small sample size is also reported as a limitation, directly impacting the results of the studies due to the greater possibility of type II error.

In this review, the findings found in the individual studies were not reflected in the overall results. This can be explained by the marked heterogeneity of the primary studies, which had different proposed intervention protocols. Following the current findings, Meng et al. (2020) [10] analyzed the effects of muscle-strengthening exercises in three subgroups of individuals with ALS, finding no significant differences in the outcome of peripheral muscle strength compared to the group without exercise or with usual care.

Contrary to these findings, Rahmati and Malakoutinia (2021) [28], in a systematic review, carried out comparative analyses between three subgroups performing aerobic exercises, resistance exercises, and combined exercises (aerobic and resistance) throughout 5–24 weeks. The authors found that combined training has a positive effect on increasing upper and lower limb muscle strength in individuals with ALS.

The divergent findings between the previously mentioned study [28] and the present review may be related to the need for more standardization regarding the instruments used to assess the primary outcome of muscle strength in the present study. This review included data on assessment using subjective measures, such as the manual muscle strength test, and objective measures and the gold standard for this type of assessment, such as isokinetic and manual dynamometers. Due to this lack of systematic approach, greater variability was observed in the results, which differed from the previous review [28]. In addition, the previous review [28] considered the assessment of upper and lower limb muscle strength separately, and categorization into these subgroups may have led to greater homogeneity and consequently less varied results.

Similar to the findings of Rahmati and Malakoutinia (2021) [28], Liu et al. (2009) [29] observed that individuals with ALS who exercised and were in the early stages of the disease obtained better results in terms of muscle strength when compared to those who were in the early stages but did not exercise. In this review, it was not possible to cluster the data according to disease status, although this analysis had been planned. Sorting by disease stage could likely reveal different effectiveness, since it directly influences the functional and motor characteristics, such as the ability to produce muscle strength. Thus, individuals in the early stages may not show muscle strength impairments as significant as those in the moderate to advanced stages of the disease [30].

Meanwhile, the meta-analyses also showed no significant effect in favor of the experimental interventions in terms of functionality and fatigue. Despite this observation, Rahmati and Malakoutinia (2021) [28], when carrying out a meta-analysis comparing the effects of physical exercise on the functional capacity of ALS patients, found statistical significance in favor of the intervention group. The subgroup analysis according to the type of exercise detected a significant effect as to the benefit of the intervention group only in the combined exercises (aerobic and resistance). It is important to note that, while the authors considered all types of experimental intervention aimed at improving health or physical conditioning, they included 16 studies in the analysis [26], in this review, the interventions were specifically aimed at muscle strengthening, a topic that is still little covered in the literature, which resulted in only eight studies being included.

Furthermore, Ortega-Hombrados et al. (2021) [11] analyzed, in a systematic review, the effects of an exercise program in people with ALS, using the ALSFRS-r. The study addressed groups that were exposed to therapeutic interventions, including aerobic, strengthening, and muscle resistance exercises, and compared the short-, medium- and long-term effects with control interventions. It was found that the exercises practiced over one month (short term) did not show clear effects and that the hypertrophy caused by the exercises should only appear after this period. In the medium term, it was seen that the weekly frequency of training was an important variable to consider since individuals who trained twice a week had better results than those who trained five times a week, reinforcing the need to balance exercises to avoid the processes of disuse and overuse. In the long term, it was found that individuals who underwent therapeutic exercises had better functional scores, as well as a lower rate of falls than those with a sedentary lifestyle, demonstrating the importance of longer interventions to increase the functionality of people with ALS [11]. Since the studies included in this review only presented data for the short-term evaluation, this may have influenced the results, so as not to verify the effectiveness of the outcomes analyzed, such as functionality.

On the fatigue issue, it is worth mentioning that its appearance is not only linked to muscular conditions, such as disorders in muscle activation or muscular abnormalities due to disuse. This symptom is also associated with cardiorespiratory deconditioning and psychological factors [20] and is therefore multifactorial. Therefore, muscle training alone is not a factor capable of reversing or mitigating fatigue in this population. Previous systematic reviews addressing muscle strengthening in this group did not find positive effects on this outcome either, which limits more assertive conclusions in this regard [10,11,28–30].

Although fatigue is frequently reported by individuals with ALS, it remains among the least treated symptoms in this population [31]. Moreover, related factors, e.g., malnutrition, respiratory problems, medication, and depression, can contribute to the exacerbation of the perception of fatigue [32]. This is one of the symptoms that tends to take on a chronic pattern in this population, due to the combination of physical and psychological aspects associated with its onset [20]. Among the studies included in the quantitative analysis of fatigue, only one observed a significant improvement in the participants allocated to the experimental intervention. However, in the rest of the studies, the fatigue reported did not exceed the expectations of exacerbation related to the progression of the disease itself.

Exercise interventions are relatively safe; however, adverse events may arise [33], and are generally related to complaints of exacerbation of pain, excessive fatigue, myalgia, and, to a lesser extent, cardiorespiratory alterations such as arrhythmias and respiratory discomfort. Among the studies included in this review, no serious adverse events were reported. The occurrence of adverse events is known to lead to delays in carrying out the intervention proposed by the study, lack of safety for the participant in the treatment, reduced adherence, and generated financial costs for individuals. Therefore, it is crucial to constantly monitor participants' responses to the proposed interventions to minimize the risk of serious adverse events [34].

In this review, the evidence quality was classified between low or very low, due to the high heterogeneity level, small sample size, wide confidence intervals, and inadequate methodological criteria in the included studies. In fact, three studies did not reach the minimum score to categorize methodological rigor as adequate, with reservations linked mainly to the allocation process and blinding of those involved in the study. In systematic review studies, every effort should be taken to ensure the methodological rigor and quality of the evidence included, since low-quality evidence and poorly conducted studies can lead to recommendations that do not reach a level of quality sufficient to support clinical decision-making and the development of health policies.

Despite the lack of robust evidence with adequate methodological rigor, it is known that the practice of physical exercise, when properly prescribed, becomes a key factor in the management of the impairments characteristic of ALS [31]. In this study, although the experimental interventions did not demonstrate superiority concerning the control interventions, it can be considered feasible to carry out muscle-strengthening interventions in individuals with ALS, since no harm associated with this practice was identified. However, there is a clear need for more randomized clinical studies to evaluate the effects of motor interventions for muscle strengthening in individuals with ALS, with standardized evaluation instruments and medium and long-term evaluations, with less risk of bias caused by inadequate masking of participants and evaluators and lack of clarity in the data reported in the study. It is known, however, that this type of study is very challenging to carry out with this population, due to the progressive nature of the disease and the short life expectancy, as well as the diversity of clinical phenotypes of the affected individuals [35]. Moreover, age is known to be a crucial factor in the progression of ALS and also directly influences the outcomes of peripheral muscle strengthening interventions. In the studies included in this review, no significant discrepancies were observed in the age range, which varied between 50 and 62 years. Additionally, a subgroup analysis was conducted considering studies with participants younger and older than 60 years, and no statistically significant differences were found.

Due to the aforementioned aspects and considering the scarcity of results in the literature regarding muscle strengthening in individuals with ALS, a subgroup analysis was planned and conducted to compare strengthening interventions that used body weight/functional exercises as resistance versus those that incorporated external loads, such as dumbbells, ankle weights, among others, in order to identify whether there were differences between these types of interventions. On the other hand, an important aspect regarding the impact of gender differences on the results was not included in the protocol, and no subgroup analysis was added in this review. This is an important factor to be examined; however, it is known that clinical studies involving individuals with ALS typically have small sample sizes compared to other neurological conditions, and results are not usually reported separately by gender—an aspect that limits data analysis considering gender differences.

This review was conducted following PRISMA guidelines (see S5 File) and following a registered and previously published protocol. An important feature of this study is the evaluation of various physiotherapeutic interventions aimed at peripheral muscle strengthening in individuals with ALS, being the only review to exclusively evaluate this type of intervention concerning the outcomes considered. The different muscle strengthening interventions found can reflect the diversity of approaches that can be adopted and highlight the lack of a standardized protocol in the literature for this purpose in this specific population. The search strategy applied was comprehensive, the selection of studies was careful, and the gray literature was scoured so that no potential studies were excluded. The fact that clinical trials and quasi-experimental studies were included may have directly influenced the assessment of the quality of the evidence and the meta-analyses; however, due to the difficulty of conducting studies with this population, we consider it prudent to consider different types of studies.

## Protocol changes for revision

To provide a better outcome, it was necessary to apply some changes to the original protocol. These amendments were made to cover new data and allow more studies to be included. The main changes are described below:

1.  Evaluation tools

1.1.   Peripheral muscle strength disclosure

The original protocol stipulated that peripheral muscle strength would be assessed using the manual strength test, muscle strength grading scale, isokinetic, or manual dynamometer. However, while reviewing and analyzing the potentially eligible studies, it emerged that the outcome in question was measured by three other methods not initially foreseen: the 1-repetition maximum test, the 30-second sit-and-stand test, and the 5-repetition sit-and-stand test. Considering the small number of papers for selection, the data obtained from these tests was analyzed and added to the review and meta-analysis, since they are also valid, safe, and functional measures for assessing lower limb muscle strength [36–38].

1.2.   Fatigue Disclosure

In the original protocol, it was planned that fatigue would only be assessed by the FHS. Subsequently, it became necessary to add data assessed using the fatigue subscale of the CIS instrument, which has satisfactory psychometric properties and assesses four domains of fatigue: the subjective experience of fatigue, reduction in motivation, reduction in activity, and reduction in concentration [39].

## Conclusion

Although an individual descriptive analysis of the primary studies indicated the superiority of the experimental intervention compared to the control, the physiotherapeutic interventions for peripheral muscle strengthening were not significantly effective in individuals with ALS when evaluated immediately after the intervention was completed. The lack of serious adverse events, on the other hand, reflects a favorable point for this type of intervention. The wide confidence intervals, heterogeneity of the studies, small sample numbers, and methodological flaws limit the level of confidence in the evidence. As this is the first systematic review dealing exclusively with the effects of physiotherapeutic interventions for peripheral muscle strengthening in ALS, there is a need for a greater number of studies with higher methodological rigor to assess the short-, medium- and long-term impacts of this intervention.

## Supporting information

**S1 File.  Search strategy.**
(DOCX)

**S2 File.  Data extraction form.**
(DOCX)

**S3 File.  Included studies information.**
(XLSX)

**S4 File.  Data extraction table.**
(XLSX)

**S5 File.  PRISMA checklist.**
(DOCX)

## Acknowledgements

The authors thank the Laboratory for Technological Innovation in Health (LAIS/HUOL) at Federal University of Rio Grande do Norte (UFRN).

## Author contributions

**Conceptualization:** Aline Alves de Souza, Karen de Medeiros Pondofe, Lorenna Raquel Dantas de Macedo, Tatiana Souza Ribeiro.

**Funding acquisition:** Ricardo Alexsandro de Medeiros Valentim.

**Investigation:** Aline Alves de Souza, Stephano Tomaz da Silva, Amanda Mayra Pereira Régis, Diogo Neres Aires.

**Methodology:** Aline Alves de Souza, Stephano Tomaz da Silva, Amanda Mayra Pereira Régis, Karen de Medeiros Pondofe, Luciana Protásio de Melo, Ana Raquel Rodrigues Lindquist, Lorenna Raquel Dantas de Macedo, Tatiana Souza Ribeiro.

**Project administration:** Tatiana Souza Ribeiro.

**Writing – original draft:** Aline Alves de Souza.

**Writing – review & editing:** Aline Alves de Souza, Luciana Protásio de Melo, Ana Raquel Rodrigues Lindquist, Lorenna Raquel Dantas de Macedo, Tatiana Souza Ribeiro.

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
