## [Decision Letter · Decision Letter 0]

27 Dec 2024

PONE-D-24-42912Muscle strengthening in individuals with Amyotrophic Lateral Sclerosis: a systematic review with meta-analysesPLOS ONE

Dear Dr. Souza,

Thank you for submitting your manuscript to PLOS ONE. After careful consideration, we feel that it has merit but does not fully meet PLOS ONE’s publication criteria as it currently stands. Therefore, we invite you to submit a revised version of the manuscript that addresses the points raised during the review process.

We look forward to receiving your revised manuscript.

Kind regards,

Masoud Rahmati

Academic Editor

PLOS ONE

“This study was supported by the Coordenação de Aperfeiçoamento de Pessoal de Nível Superior – Brazil (CAPES) – Finance Code 001, and by the National Council for Scientific and Technological Development and Ministry of Health, through a Decentralized Execution Term (TED 132/2018).”

3. We note that your Data Availability Statement is currently as follows: [All relevant data are within the manuscript and its Supporting Information files]

5. As required by our policy on Data Availability, please ensure your manuscript or supplementary information includes the following:

Reviewers' comments:

Reviewer's Responses to Questions

**Comments to the Author**

1. Is the manuscript technically sound, and do the data support the conclusions?

Reviewer #1: Partly

2. Has the statistical analysis been performed appropriately and rigorously?

Reviewer #1: Yes

3. Have the authors made all data underlying the findings in their manuscript fully available?

Reviewer #1: Yes

4. Is the manuscript presented in an intelligible fashion and written in standard English?

Reviewer #1: Yes

5. Review Comments to the Author

Reviewer #1: Dear Authors

I have read your manuscript in which you have investigated the effectiveness of Muscle strengthening in individuals with Amyotrophic Lateral Sclerosis. The paper is well written and provides important information about individuals with ALS. Nevertheless, there are some considerations need to be addressed here.

• Could gender have an impact on the results? And why aren’t gender differences thoroughly addressed in the discussion?

• Given the significant variations in duration of interventions, and other key factors in previous studies, does it raise concerns about the foundation of your research?

• The aforementioned concern is also associated with the discrepancy in intervention of previous researches.

• Why have exercises like dumbbells and electrostimulation exercises been studied together?

• Since increasing age can affect strength training results, doesn't these age differences in previous studies affect the results of your study?

6. PLOS authors have the option to publish the peer review history of their article (what does this mean? ). If published, this will include your full peer review and any attached files.

**Do you want your identity to be public for this peer review?** For information about this choice, including consent withdrawal, please see our Privacy Policy .

Reviewer #1: No

---

## [Author Response · Author response to Decision Letter 1]

14 Feb 2025

First of all, we would like to thank the reviewer for his availability and for his very pertinent comments, which helped us to improve our work and clarify essential issues within the research. Below you will find the questions raised by the reviewer and the answers to all of them, with clarifications and indications of the changes made to the text.

• Could gender have an impact on the results? And why aren’t gender differences thoroughly addressed in the discussion?

In fact, gender differences in muscle strengthening treatments for people with ALS should be considered to optimize results. Individualized interventions that consider biological, clinical, and social factors can offer more significant benefits and improve patients' quality of life; however, this is not a reality found in the scientific evidence available to date. Regarding the studies included in our review, all selected references had their samples composed of both genders, making it impossible to perform a more in-depth analysis of the impact of gender differences on the outcomes evaluated.

We appreciate this pertinent comment and have included a paragraph in the discussion addressing this topic in the discussion (paragraph 15 of the discussion, lines 5 to 10).

• Given the significant variations in duration of interventions, and other key factors in previous studies, does it raise concerns about the foundation of your research?

The heterogeneity of the primary studies, which includes the different interventions applied and their durations, in fact, limited the level of confidence in the results of our systematic review. However, this was well specified in the discussion (paragraph 14, lines 6 and 12; paragraph 16, lines 7 to 11) and conclusion of the present study.

However, this same concern consists of an important result, which highlights the need for more robust studies related to peripheral muscle strength in ALS, which present better standardization in the evaluation of the outcome, in the characteristics of the interventions and in the presentation of data, so that, in this way, new analyses can be performed and the levels of evidence and confidence can be increased.

• The aforementioned concern is also associated with the discrepancy in intervention of previous researches.

Exactly. As previously stated, the heterogeneity of the studies was also verified in terms of discrepant interventions applied. It is likely that the different interventions for muscle strengthening in the included studies reflect the diversity of approaches currently adopted for individuals with ALS, which, in turn, may be linked to the lack of a standardized protocol in the literature for this purpose, for this population. It is important to emphasize, however, that in the present review additional analyses (or subanalyses) were performed, grouping studies with similar characteristics, in an attempt to greater homogenize the sample, allowing a more precise evaluation of the effects of the interventions on the outcomes. (paragraph 16 of the discussion, lines 2 to 7)

• Why have exercises like dumbbells and electrostimulation exercises been studied together?

We sought to include different types of interventions because this is a topic that has been little discussed in the literature to date, with scarce and sometimes controversial results. For this reason, we included several possibilities for muscle strengthening, using accessories, techniques and devices, in order to broaden the search range of primary studies that investigated peripheral muscle strengthening in this population.

Aware of the heterogeneity that this could cause, we considered conducting additional analyses with subgroups that performed strengthening with body weight/functional exercises or that used external loads, such as dumbbells, ankle weights, etc., in order to identify whether there was a difference between these interventions. Comments on this analysis are in paragraph 15 of the discussion, lines 1 to 5.

• Since increasing age can affect strength training results, doesn't these age differences in previous studies affect the results of your study?

Age is a crucial factor in the progression of ALS and directly impacts the results of exercise interventions, with studies suggesting that younger individuals with ALS may benefit more from muscle strengthening interventions due to greater muscle reserve and neural plasticity. In the studies included in this review, the mean age range of participants was assessed and, following the natural course of development of the disease, where individuals are often diagnosed after the age of 60, the age range varied between 50 and 62, with no significant discrepancies in the age range observed in the included studies. In addition, we included a subgroup analysis considering the studies with participants under and over 60 years old, and no statistically significant differences were found, as demonstrated in the results and discussed in paragraph 14 of the discussion, lines 12 to 17.

---

## [Decision Letter · Decision Letter 1]

25 Feb 2025

Muscle strengthening in individuals with Amyotrophic Lateral Sclerosis: a systematic review with meta-analyses

PONE-D-24-42912R1

Dear Dr. Souza,

We’re pleased to inform you that your manuscript has been judged scientifically suitable for publication and will be formally accepted for publication once it meets all outstanding technical requirements.

Kind regards,

Masoud Rahmati

Academic Editor

PLOS ONE

Additional Editor Comments (optional):

Reviewers' comments:

Reviewer's Responses to Questions

**Comments to the Author**

1. If the authors have adequately addressed your comments raised in a previous round of review and you feel that this manuscript is now acceptable for publication, you may indicate that here to bypass the “Comments to the Author” section, enter your conflict of interest statement in the “Confidential to Editor” section, and submit your "Accept" recommendation.

Reviewer #1: All comments have been addressed

2. Is the manuscript technically sound, and do the data support the conclusions?

Reviewer #1: (No Response)

3. Has the statistical analysis been performed appropriately and rigorously?

Reviewer #1: (No Response)

4. Have the authors made all data underlying the findings in their manuscript fully available?

Reviewer #1: (No Response)

5. Is the manuscript presented in an intelligible fashion and written in standard English?

Reviewer #1: (No Response)

6. Review Comments to the Author

Reviewer #1: (No Response)

7. PLOS authors have the option to publish the peer review history of their article (what does this mean? ). If published, this will include your full peer review and any attached files.

**Do you want your identity to be public for this peer review?** For information about this choice, including consent withdrawal, please see our Privacy Policy .

Reviewer #1: No

---

## [Editor Report · Acceptance letter]

PONE-D-24-42912R1

PLOS ONE

Dear Dr. Souza,

I'm pleased to inform you that your manuscript has been deemed suitable for publication in PLOS ONE. Congratulations! Your manuscript is now being handed over to our production team.

Kind regards,

on behalf of

Dr. Masoud Rahmati

Academic Editor

PLOS ONE